# Human Digital Twin in Industry 5.0: A Holistic Approach to Worker Safety and Well-Being through Advanced AI and Emotional Analytics

**DOI:** 10.3390/s24020655

**Published:** 2024-01-19

**Authors:** Saul Davila-Gonzalez, Sergio Martin

**Affiliations:** 1Escuela Internacional de Doctorado, Universidad Nacional de Educación a Distancia (UNED), 28040 Madrid, Spain; sdavila30@alumno.uned.es; 2Industrial Engineering Faculty, Universidad Nacional de Educación a Distancia (UNED), 28040 Madrid, Spain

**Keywords:** Human Digital Twin, Industry 5.0, Internet of Things, artificial intelligence, health and safety environment, physiological health, mental health, emotional well-being

## Abstract

This research introduces a conceptual framework designed to enhance worker safety and well-being in industrial environments, such as oil and gas construction plants, by leveraging Human Digital Twin (HDT) cutting-edge technologies and advanced artificial intelligence (AI) techniques. At its core, this study is in the developmental phase, aiming to create an integrated system that could enable real-time monitoring and analysis of the physical, mental, and emotional states of workers. It provides valuable insights into the impact of Digital Twins (DT) technology and its role in Industry 5.0. With the development of a chatbot trained as an empathic evaluator that analyses emotions expressed in written conversations using natural language processing (NLP); video logs capable of extracting emotions through facial expressions and speech analysis; and personality tests, this research intends to obtain a deeper understanding of workers’ psychological characteristics and stress levels. This innovative approach might enable the identification of stress, anxiety, or other emotional factors that may affect worker safety. Whilst this study does not encompass a case study or an application in a real-world setting, it lays the groundwork for the future implementation of these technologies. The insights derived from this research are intended to inform the development of practical applications aimed at creating safer work environments.

## 1. Introduction

This research seeks to enhance human health and safety at work while keeping worker productivity at high rates. It proposes collecting data on mental, emotional, and physical aspects of human behavior using various technologies for data sensing and analysis. It uses AI models for data aggregation with the purpose of assessing workers’ health and safety, not only in the short term but also in the long term, as well as preventing stress-related issues like anxiety or depression by creating a “Human Digital Twin Profile”.

The importance of addressing stress and burnout in the workplace through cutting-edge technologies for creating a better and more accurate Human Digital Twin (HDT) is emphasized in this work. The concept of emotion artificial intelligence (EAI) is introduced; it offers a view through which it can help to detect and prevent critical and fatal situations. This concept investigates how AI systems can recognize and analyze human emotions to trigger the necessary corporate mechanisms to avoid mental health problems like anxiety and depression before it is too late.

The motivation for this work stems from the significant economic impact of stress-related issues, such as anxiety and depression, that the World Health Organization (WHO) published in 2022 [1], with billions of workdays and trillions of dollars lost for worldwide companies. Based on this impact, the zero-accidents goal is currently a major concern in almost every large-scale project, but its efficiency is constantly evolving. This work aims to provide a deep understanding of how cutting-edge technologies can help to achieve this goal.

This research explores the field of smart monitoring and AI integration for worker safety and well-being in industrial environments, mainly focused on oil and gas construction plants which are predominantly outdoor scenarios. Firstly, the study explores the latest advancements in smart monitoring devices, particularly those that focus on the physical body. A significant objective is the integration of a smart weight scale into the DT framework to offer a comprehensive insight into worker’s body composition and potential health risks. Furthermore, this study aims to incorporate a smartwatch that would serve multiple purposes, like GPS for tracking, heart rate variation (HRV) monitoring to identify stress levels, or evaluating sleep quality, all in real-time, to ensure timely safety alerts. Concerning the mental body, to recognize the importance of mental well-being in the workplace, this study also incorporates personality tests and their analysis. These tests provide a deeper understanding of workers’ mental mindsets and potential stressors. Lastly, to identify the emotional body, this work investigates the power of AI to collect emotional status through several mechanisms. Throughout video logs, emotions can be recognized from facial expressions and speech-to-emotion algorithms. In the ChatGPT conversational chatbot, emotions can be recognized from text-to-emotion analytics.

This paper offers a structured exploration of the research topic. Beginning with this introduction, it moves to the Section 2, which encompasses the current state of the art, including the relevance of Industry 5.0 and Digital Twins and the importance of the Health and Safety Environment (HSE). The development segment in Section 3 delves into the details of the user profile, and mental, emotional, and physical bodies. Pertinent legal considerations are outlined in Section 4, as well as the conclusions drawn and potential avenues for future studies. Figure 1 illustrates the high-level architecture of this work, encompassing three primary domains: physical, mental, and emotional. Each domain is expanded with its respective components and subcomponents, showcasing the comprehensive approach to the HDT model proposed.

Given the paper’s comprehensive exploration of Industry 5.0, Digital Twins, and their intersection with HSE protocols, the potential audience is multifaceted. It includes industry professionals interested in integrating technological advancements for employee well-being, researchers focused on DT technology and its practical applications, and officials interested in understanding the legal and ethical implications of these technologies. Additionally, HSE officers, aiming to leverage modern tools for better workplace practices, will find the insights particularly valuable.

This work makes a significant contribution to the state-of-the-art in Human Digital Twins (HDT) within health and safety environments. The main contributions of the research are described as follows.

Real-time monitoring: Implementing an integrated system for real-time monitoring and analysis of workers’ physical, emotional, and cognitive aspects.Advanced emotional analysis: Employing AI and natural language processing for emotional analysis in written communications and video logs.Cognitive analysis through personality tests: Utilizing personality tests to assess cognitive aspects related to emotions.Proactive interventions: Facilitating proactive interventions for safer work environments via holistic monitoring and AI analysis of physical, emotional, and cognitive factors.Integration of cutting-edge technologies: Applying Digital Twin technologies and advanced AI to enhance worker safety and well-being in industrial settings. It also includes the use of smartwatches and smart weight scales.

## 2. Materials and Methods

This section details the conceptual framework and strategic approach utilized in the research. It outlines the proposal of a potential framework designed to integrate various innovative technologies. It explains the methods used in capturing and analyzing diverse human metrics, emphasizing how these methods contribute to enhancing safety and well-being in industrial work environments. Further details on the specific steps followed in this research are explained below.

State-of-the-art analysis. The paper extends its state-of-the-art analysis beyond Digital Twin and Industry 5.0, encompassing advanced sensor technologies, AI for emotional and cognitive assessment, and their application in industrial environments. It explores the integration of these technologies to mitigate work-related stress, particularly in high-stress settings like oil and gas construction plants, addressing potential mental health issues such as anxiety and depression, as referenced in [2]. This comprehensive approach ensures a detailed understanding of the interplay between these technologies and their impact on worker well-being.The design and development of an HDT conceptual framework for Industry 5.0. It involved:
Web platform development. The platform intricately captures a range of human metrics using advanced algorithms. Physical data is sourced from devices like smartwatches and weight scales, mental metrics through personality tests, and emotional aspects via video logs and a chatbot for emotional analysis. These are integrated into a Human Digital Twin profile and collated into a single JSON file. Additionally, the platform features a user-friendly dashboard that displays the risk level of each parameter, providing a comprehensive view of workplace safety and worker well-being. The backend system is powered by Google Firebase (v. 9.15.0)*,* ensuring robust data management, while the frontend utilizes the Angular framework version 14.Integration of several commercial sensors. To ensure accurate data collection, smart sensors, including a Garmin smartwatch and a Xiaomi smart weight scale, are integrated into the system backend using some of the Google Firebase components, like Firebase Functions and Firestore Database, and presented in the frontend using Angular Material Design and third-party libraries to generate data graphically. These devices feed real-time data into the platform, which uses AI analytics to assess the outcomes. The web platform gathers the emotional state of workers through video logs, by facial expressions and speech emotion recognition, and personality tests.Integration with APIs. The platform leverages API connections for comprehensive data integration. To access physical metrics, the Garmin Developers Portal is used through a push REST API, enabling real-time data synchronization from Garmin devices. A similar approach is used for the smart weight scale, through which data is pushed to a service created in Firebase Functions. The integration with the One AI platform and ChatGPT from OpenAI, both through REST services, enhances the platform’s cognitive and emotional analysis capabilities. Additionally, the Sentino API is utilized for personality tests, seamlessly integrating these insights into the platform. This multi-faceted API integration is crucial for the AI component of the platform to proactively identify potential risks and enable timely preventive measures.


## 3. Sate of the Art

### 3.1. Industry 5.0 and Digital Twins

Industry 5.0 with its basis on human-centric and collaborative work environments and within the scope of this research, enables a transformative approach to create safer and more efficient workplaces [3]. Built on the foundation of Industry 4.0, which emphasizes automation and interconnectivity, Industry 5.0 integrates the human element more profoundly, prioritizing worker well-being and sustainability [4]. Advanced robots and automation tools are no longer seen merely as replacements but as collaborative partners that can operate alongside humans to minimize risks and hazards. These intelligent systems can predict potential safety breaches and adjust operations in real time, ensuring a proactive approach to safety workplaces [5].

Furthermore, with the rise of wearable technologies and sensors, there is an increased capacity to monitor workers’ physical and mental health in real-time. These devices can track vital signs, fatigue levels, and even emotional stress, offering immediate insights and allowing for timely interventions to prevent health-related issues. This not only ensures the physical well-being of workers but also addresses potential mental and emotional health challenges, fostering an environment where psychological safety is paramount.

In addition, data analytics and AI play a crucial role. They aid in analytics and identification of patterns and trends related to workplace accidents or illnesses, enabling companies to refine their HSE protocols continuously. Emphasis on customization also ensures that solutions are tailored to the specific needs and challenges of each sector, rather than adopting a one-size-fits-all approach.

Digital Twins (DTs) concept aims to create digital replicas of physical objects or systems, facilitating real-time monitoring, simulation, and forecasting [6]. Extending this notion, Human Digital Twins (HDT) has emerged and is being applied across various fields, like healthcare, smart manufacturing [7], or smart construction [8,9]. HDTs are envisioned as a powerful tool to monitor, optimize, and forecast processes, thereby contributing to the continuous improvement of human well-being and quality of life [10].

Based on the research and work completed in this paper, the definition of HDT in industrial work environments, within the context of HSE, is enhanced. Workers are more than their physical parameters, as the industry has traditionally considered, and the model proposed here aims to gather within a JSON profile, physical parameters, emotional status, and cognitive trends. With this enriched HDT, a worker can be characterized on its virtual model in a more precise way and within a single profile that can be transferred to multidisciplinary areas. Concerning safety and well-being, with the approach in which all HDT data is collected in a single profile, it can be easier for professionals to analyze and establish predictions or corrective actions when needed.

Progressing further, the discourse around Digital Twins has evolved to encompass autonomous, context-aware, and adaptive AI models. These models are seen as the cornerstone in enabling companies to navigate the challenges of modern and dynamic environments [11]. The adaptive nature of HDTs is crucial for addressing challenges related to competitiveness, productivity, and sustainability, which are important for ensuring safety and well-being at work. Although the theoretical framework for the implementation of Digital Twins exists, the practical implementations are still evolving and, unfortunately, are not always publicly available.

The potential benefits of real-time monitoring and simulation offered by DTs present a promising avenue for enhancing human well-being and safety in the workplace. The blend of technological advancements with a human-centric approach characterizes the state of the art in this domain, aiming to leverage DTs to create safer and more productive work environments.

Considering the goal of this work, which is to create an efficient HDT within the scope of safety and productivity in industrial environments from a humanistic perspective in which workers physical and mental domains are well understood, none of the scientific papers currently published present a solution that involves all aspects related to safety in the industrial workplace. For instance, some papers [3,5,9,12,13,14] present a cyber-physical system solution to gather physiological data of operators and construction worker through different physical sensors, which can be analyzed through specific designed interfaces. However, none of them expands the safety scope to the relationship between worker emotional safety and well-being piloted by AI models. Other papers examine caring for the mental domain [15,16,17,18,19,20] using AI technology as these mental disorders can occasionally cause stress-related problems and have a significant impact on safety and productivity. There are also published papers that consider the importance of emotional well-being at work to achieve a safer work environment, like [21,22,23,24,25], which use different mechanisms, such as virtual reality, cameras or facial expression recognition, to gather emotional status information which is analyzed through AI models; however, they do not consider the physical cross-data, suggest a platform to analyze, or expose the users to the possibility so they can be aware of their overall situation.

This comprehensive analysis of health and safety in industrial environments underscores the urgent need for innovative solutions that address the multifaceted challenges of worker safety and well-being. While conducting this research, no papers were found with a comprehensive HDT, i.e., encompassing the three main domains of a human being within the HSE: physical, mental, and emotional. Table 1 represents the most relevant publications that have been studied which encompass some isolated aspects of this work.

### 3.2. Health and Safety Environment

In the paradigm of Industry 5.0, the HSE Department stands at the forefront, safeguarding worker well-being and fostering a secure working ambiance. As the industrial arena witnesses a shift toward a more interconnected and human-centric model, the role of HSE has evolved beyond traditional measures. This transformation emphasizes the importance of technology and worker monitoring, embedding safety and health protocols within current operations.

Human Digital Twins can combine physiological, psychological, and behavioral data to predict, analyze, and guide actions. Within this framework, the HSE Department adopts a dual role. First, it acts as a custodian of these digital entities, ensuring that they retain accurate reflections of their human counterparts, updated in real-time and treated with the utmost data integrity and confidentiality. Second, it functions as an interpreter, leveraging insights drawn from these DT to implement timely interventions, refine safety protocols, and tailor worker engagements to minimize risks.

The interplay between HSE, Industry 5.0, and Digital Twins embodies a synergy aimed at revolutionizing safety standards. With Industry 5.0 emphasizing a collaborative coexistence of humans and machines, the responsibility of HSE departments is to ensure that this collaboration is seamless and safe. Digital Twins, in this context, provide a granular view of worker health, well-being, and interaction with industrial systems, which enables HSE professionals to anticipate potential threats, simulate various safety scenarios, and innovate evidence-backed strategies. For instance, if an HDT can predict fatigue or high stress levels in a worker operating heavy machinery, immediate interventions, such as breaks or rotations, can be recommended, circumventing potential hazards [5].

## 4. Results

The developed web platform is based on an open-source front-end built on Angular framework, powered by Google. Angular, specifically designed for dynamic Single-Page Applications (SPAs), supports modular architecture, enabling streamlined, reusable coding. It employs TypeScript, an advanced JavaScript variant, introducing features like static typing and classes. This not only fortifies code organization but also simplifies debugging and maintenance.

This work makes use of the different elements of Angular’s architecture, like components (UI segments), directives (enhancing HTML behavior), services (data sharing across components), and modules (code organization). It excels in data binding, maintaining data-view synchronization, dependency injection, ensuring modular, testable code, and routing, for application navigation. These attributes have been meticulously deployed to achieve the primary goal of reinforcing workplace safety, physically, mentally, and emotionally.

Complementing the frontend, a robust backend powered by Google Firebase has been implemented. Firebase enhances web and mobile app development with its various services. This project mainly utilizes authentication, Firestore database, storage, functions, and hosting features.

The critical component of authentication, safeguards access, preserving services like OpenAI from unwarranted intrusions. Firestore serves as a flexible database solution, employing a NoSQL approach for smooth data management. Assets like videos and images are stored in Firebase’s storage. Firebase Functions offers serverless computing, bridging peripherals like the smartwatch and the smart weight scales to the database. Lastly, Firebase Hosting, coupled with a global CDN, ensures our platform remains accessible worldwide, showcasing all its groundbreaking features.

Figure 2 presents the detailed architectural depiction of front-end and back-end framework components. This figure illustrates the various elements constituting the front-end and back-end of the proposed framework, including user interfaces, data processing layers, APIs, and server-side infrastructure. Each component is annotated to indicate its role and interaction within the overall system architecture.

### 4.1. User Profile

This interface allows workers to manually enter valuable data, such as profession, work location, complexity, height, or gender (see Figure 3). This additional data is used to complete the Digital Twin profile and feeds the AI model so it can automatically calculate new prompt responses and trigger new worker status results.

The complete and updated DT profile can be found embedded in this interface in a JSON format, which is a lightweight data exchange format that is easy for humans to read and write, and easy for machines to parse and generate. JSON is based on a subset of the JavaScript programming language and this work uses it to transmit data between the server and the web application.

### 4.2. Mental Body

The emergence of Industry 5.0, with its human-centric approach, has underscored the importance of human mental health. Recognizing that certain behavioral paradigms can be threats to individual and collective well-being, this research illuminates the suggestion of personality assessments as tools for reading these intricate dynamics.

The extensive canvas of human personality, marked by plenty of traits and behaviors, offers a variety of individual peculiarities that must be guided by professionals in the area of specific AI models designed for such a purpose. Understanding this situation is essential to be able to establish correlations between certain personality traits and mental health outcomes. For instance, some research has revealed that individuals with pronounced neuroticism face heightened vulnerabilities to mental disorders, such as anxiety and depression [26]. In contrast, positive attributes like resilience and optimism often act as protective bulwarks, fostering better mental health [27].

Beyond these direct correlations, the interplay between personality and mental well-being extends to influencing coping mechanisms and shaping social affiliations. In essence, an individual’s inherent traits don’t just determine their mental health trajectory but also sculpt their broader life experiences. Incorporating these insights into workplace environments, especially in the context of Industry 5.0, can revolutionize organizational dynamics. This research enthusiastically advocates for integrating personality tests within workplace frameworks, which can help in crafting safety measures, finetuning training programs, and even delineating team interactions.

This work, with the integration of Sentino API, has enabled an interface to analyze, Figure 4, and fill, Figure 5, the personality test based on the Big Five Personality Traits or the Five Factor Model (FFM). Comprising openness, conscientiousness, extraversion, agreeableness, and neuroticism, these traits offer an in-depth probe into the human psyche, facilitating rich insights into emotional patterns, behavioral tendencies, and cognitive processes. Leveraging the capabilities of Sentino API to administer and interpret these tests provides an added layer of precision and accessibility. These insights can play a pivotal role in fostering harmonious interpersonal relationships, both in personal and professional areas.

### 4.3. Emotional Body

This research highlights the importance of integrating the emotional understanding of workers to be able to analyze and predict their response to certain stressful situations. This emotional recognition, understanding, and management, is what the psychologist Daniel Goleman coined as emotional intelligence. In Human Digital Twins, this can lead to behavioral models that understand personal reactions supporting tailored interventions, fostering effective communication in organizations, or improving decision-making to enhance workplace well-being.

From a corporate perspective, the “Emotional Salary” concept, which refers to non-financial work benefits like recognition, is gaining popularity. Given the connection between emotional well-being, stress, and potential health concerns [28,29], this research aims to assess employees’ emotional states using two methods: analyzing periodic video logs that translate facial expressions and speech into emotions, and an empathetic chatbot prompted to extract worker emotions through conversations. After the data is collected by both mechanisms, it is presented in an emotional dashboard, Figure 6, to graphically show the emotions gathered from the evaluator and the historical video logs for further professional analysis.

Video analytics are a powerful resource to extract information about many different aspects. Historically, CCTV systems were able to detect loitering or intrusions, but since the appearance of artificial intelligence, the applications have grown exponentially, allowing these systems to detect, among many other use cases, asset behavioral patterns, object tracing, people’s emotions, facial recognition, etc.

The general idea behind the concept of the video log within the scope of health and safety in industrial environments is to introduce a technique to determine the worker’s emotions through a video and audio recording. Speeches are a great mechanism to understand human emotions as both what we say and how we say it is important. Although this work only captures facial expression and speech emotions, future research can be performed to obtain an improved model of the body language of workers and the speech tone to increase the accuracy and usability of the HDT.
Facial expressions: They are basically involuntary mirrors to human emotions, often revealing feelings even before the individual is aware of them. These expressions can be detected using cameras and AI analysis to deduce an individual’s emotional reactions [19]. To ensure GDPR compliance and enhance user privacy, this project employs an in-browser model, eliminating the need to transfer images to external servers, although the video recordings are stored for testing and debugging purposes.The main tool used for facial analysis is the library “face-api”. This JavaScript face recognition API, built on the foundational TensorFlow.js, facilitates face detection, recognition, and emotional analysis in web applications. Face-api.js specializes in facial expression detection by analyzing specific facial features to predict emotions like happiness, sadness, anger, or surprise. This capability is underpinned by a pre-trained deep learning model. Initially, the system detects faces in the webcam using machine learning techniques, such as convolutional neural networks. Subsequently, the recognized faces are analyzed to determine specific emotional patterns. Figure 7 shows the video log interface and the graphical representation of the algorithm’s outcome.


Speech to emotion: It is gathered using the “webkitSpeechRecognition” library, an embedded feature in the Google Chrome browser, for speech-to-text conversion, and its output is fed to the subsequent module responsible for emotion detection. The emotion recognition from the text relies on an external API service named One AI. Their text-to-emotion feature is exceptional at discerning a worker’s emotional status during periodic video logs, safeguarding their mental, emotional, and physical well-being. Utilizing natural language processing (NLP) combined with machine learning, One AI deciphers emotions manifested in written words.Emphatic evaluator: AI-driven chatbots are tools designed to interact in a human-like way. In this study, the chatbot deployed is based on ChatGPT 3 Davinci, a model from Open AI available through API integration. Through prompt engineering, the AI model has been tailored to extract the emotions of workers through conversations. Such configurations allow the chatbot to deliver a more empathic and personalized user experience. For example, when a user conveys frustration, ChatGPT establishes a confidential space for the workers, allowing them to communicate their emotions and feelings without reservations, promoting a more inclusive work environment. These capabilities enable an in-depth analysis of workers’ emotional content, assisting employers in identifying emotions like stress or anxiety. As a result, proactive steps, such as workload adjustments or access to mental health resources, can be undertaken.

At the heart of these advanced chatbot capabilities are two pivotal AI components: machine learning and NLP. Machine learning, especially its subset deep learning, relies on layered algorithms (artificial neural networks) to discern patterns from data, driving the chatbot’s response mechanism. In parallel, NLP is instrumental in decoding user inputs, helping the chatbot identify emotional undertones, intentions, and other hidden semantics in a user’s statements [20]. Figure 8 shows the interface of the empathetic evaluator chatbot developed in Angular framework to capture and analyze workers emotions in real-time.

### 4.4. Physical Body

For HSE in industrial environments, the importance of workers being fit enough cannot be overstated. It not only ensures efficient task execution but also reduces the potential for accidents and injuries, especially for those workers who are exposed to extreme physical work. Being physically optimal allows workers to exhibit the necessary strength, agility, and endurance to effectively handle machinery, lift objects, and traverse potentially dangerous zones. Ignoring physical well-being can lead to detrimental outcomes. Fatigue, impaired cognition, and physical inadequacies may compromise decision-making and response times. To avoid such scenarios, it’s imperative for both employers and workers to prioritize physical health. This involves integrating regular exercise, balanced nutrition, and sufficient rest into daily routines. Additionally, workplace structures and protocols should be ergonomically designed, factoring in workers’ physical capabilities.

This study offers technological tools, such as smartwatches and a smart weight scale integration, to assess a worker’s physical status. The smart weight scales, equipped with bioelectrical impedance analysis, obtain weight measurements to analyze body composition metrics, including body fat, muscle mass, hydration levels, or BMI. On the other hand, smartwatches serve dual purposes: using GPS to track workers’ movements within industrial zones to ensure safety compliance, especially in high-risk areas, and monitoring critical health parameters like heart rate, stress levels, or sleep quality. Such real-time monitoring acts as a proactive defense against health threats, prompting timely interventions [14].

Body composition

The smart weight scale used in this study is the Xiaomi Weight Scale, which offers comprehensive body composition monitoring. Parameters such as weight, muscle mass, bone mass, body fat percentage, visceral fat, and basal metabolism are meticulously captured, making it an indispensable tool for proactive safety management in industrial settings.

Weight fluctuations, as a baseline measure, can indicate potential health issues. Meanwhile, muscle and bone mass measurements offer insights into physical resilience and potential vulnerabilities, critical for physically demanding environments. For instance, early detection of muscle wasting indicative of conditions like cancer cachexia is possible through muscle mass monitoring, allowing for timely interventions [30].

Body fat percentage and visceral fat measurements shed light on overall health and metabolic risks, with excessive visceral fat being a precursor to chronic diseases. In parallel, understanding basal metabolism, which indicates the caloric requirements at rest, is essential for gauging energy needs, ensuring workers are well-prepared for demanding tasks, and mitigating fatigue-induced risks.

The Xiaomi Weight Scale’s readings are extracted via the “mi-scale-exporter” Android App available on Google Play. It uses Bluetooth 5.0 to connect to the weight scale and extract the data, which it converts into human-readable form and pushes to a Firebase Function, as shown in Figure 9. Moreover, the function inserts the data into the Firebase database to be consumed later by the front-end (Figure 10). Figure 11 presents the interface developed to show weight-related information gathered from the complete physical body system.

Location

To ensure the safety and well-being of workers in industrial environments, accurately tracking their physical location to enhance safety is mandatory. In high-risk environments where many hazards are present, the time response in an emergency is essential. Additionally, not all workers possess the same training to access the different areas, so GPS location can prevent non-trained workers from entering specific hazardous areas or operating close to specific heavy equipment. The Garmin backend ecosystem acts as a necessary conduit for data transfer. First, the data from the Forerunner 45 is synchronized with the Garmin Android application; from there, it is subsequently relayed to Garmin’s servers. Then, Garmin offers a mechanism to push the data to a specific endpoint, which, in the case of this study is being completed through a Firebase function. This data is finally kept in the Firestore database, ensuring a cohesive and streamlined backend system dedicated to the overarching goal of improving worker safety.

As part of the scope of this research, an interface for location visualization with trips and alerts has been developed. Figure 12 presents an oil and gas industrial environment in which the worker positions, noted as blue markers, indicate the access to a restricted area, noted as the red area, and the system indicates that the worker might be exposed to a hazardous risk.

Heart-related parameters

This research introduces an additional technological solution to enhance the safety and well-being of workers in industrial environments based on monitoring heart-related parameters, like heart rate, sleep quality, and stress levels, through the same smartwatch that is collecting real-time positions by GPS. Real-time heart rate monitoring offers alerts for detected irregularities, enabling timely intervention for potential cardiac issues. Sleep monitoring assesses workers’ rest quality and duration, which is crucial for identifying fatigue, which affects job performance. Stress measurements, through advanced algorithms, evaluate mental and physical health, facilitating interventions like stress reduction programs.

The smartwatch used in this research is the Garmin Forerunner 45, which has built-in sensors to track heart rate and HRV data. This data is processed on the edge to deduce the user’s stress response. High HRV signifies a relaxed state, whereas low HRV indicates stress [31]. Smartwatches process HRV with considerations like respiration rate to give a comprehensive stress assessment, typically as a stress score or visual representation. Besides stress, which is currently a big concern for the WHO, there are many applications in which HRV can be a powerful tool for deducing emotional intelligence and spotting potential emotional disorders [24]. HRV can also detect disorders like anxiety or PTSD based on specific patterns, prompting individuals to seek timely treatment [25].

As in the location-focused section, heart parameters are derived from the smartwatch Forerunner 45, synced with the Garmin Android application and Garmin servers. The data then flows to Firebase Firestore via SDK. Two Firebase functions retrieve and store smartwatch data: “garminActivityDetailCallback” for positions, BPM, and stress, and “garminSleepCallback” for sleep metrics, ultimately influencing worker energy analysis. All parameters feed into an AI-powered dashboard assessing worker risk. This work also enables a dashboard to display heart rate (max and min), sleep quality metrics, calories burnt during the day, and stress levels, featuring search filters and manual date ordering (Figure 13).

### 4.5. Dashboard

The outcome of this study has culminated in an innovative dashboard that reflects a worker’s overall status using specifically chosen key performance indicators (KPIs) and artificial intelligence analysis. These KPIs, addressing the foundational aspects of a human being—mental, emotional, and physical—are integrated and dynamically updated within a Human Digital Twin profile. This profile stands as a testament to the real-time dynamism and adaptability of this system.

An important component driving the efficacy of this system is the integration of prompt engineering in the OpenAI Model Davinci 3 with the Human Digital Twin profile to evaluate the worker’s current state. By leveraging the vast capabilities of large language models (LLM), such as GPT-3, this work has carefully designed prompts that take into consideration the information available in the Digital Twin profile. The physical body group considers, as the main component, the energy (heart-related data, sleep quality, and stress identifiers gathered from the smartwatch), as illustrated in the prompt of Figure 14.

The scale light indicator is obtained by prompting the body composition obtained from the smart weight scale, as Figure 15 illustrates, and location by simply analyzing the GPS positions received from the Garmin Smartwatch (Garmin store in Madrid, Spain). A smartwatch manufactured by Garmin Ltd., and sourced in Madrid, Spain, for this research.

The emotional group shows the emotional light indicator from the video logs which comprises two retrieval mechanisms: face expressions and text to emotion. The prompt used to obtain and feed the dashboard is represented in Figure 16.

The last indicator represents the emotional status of the worker, but this time, the feeding mechanism is the emphatic evaluator and is used through ChatGPT conversations and text to emotions. The prompt used is represented in Figure 17.

The representation of the overall worker’s status has been presented using a traffic light representation, and it is obtained from GPT-3′s interpretation of the prompts mentioned. In the traffic light representation system, the color “Red” means that a worker might not be in an optimal state for the day’s tasks, while “Yellow” indicates a need for caution, suggesting that the worker can participate but might require periodic oversight or assistance. On the other hand, a “Green” signal denotes that the worker is physically or emotionally capable of performing their responsibilities. Figure 18 shows the look and feel of this mentioned dashboard, with not just light indicators but also the comments on the decision taken by the AI.

The ability of the system to generate this traffic light output, rooted in the comprehensive data from the Digital Twin profile, underscores the importance of real-time updates. As the profile is continually refreshed with current data, the dashboard remains consistently accurate and relevant, allowing for informed, timely decisions. In essence, the seamless synergy between prompt engineering, GPT-3′s capabilities, and the dynamic nature of the Digital Twin profile ensures unparalleled accuracy and currency in worker assessment.

## 5. Discussion and Conclusions

This research effectively integrates different mechanisms to assess workers’ physical, mental, and emotional well-being. Rooted in the paradigms of Industry 5.0 and Society 5.0, this work underscores the potential of smart monitoring and AI within the Digital Twin framework to enhance decision-making and improve safety at work.

Smart monitoring devices, such as a smart weight scale and a smartwatch, are integrated into the Human Digital Twin framework to control the body composition, real-time location, stress levels, and sleep quality to enhance worker physical safety in industrial environments. The work enables an interface for worker personality analysis through a Big Five test, which helps to identify individual risks and their potential advanced mitigation. The emotional profile is gathered through facial expressions, speech emotion recognition, and conversational methods. The system employs AI models to detect signs of stress, fatigue, or potential mental and emotional complications, ensuring timely support for workers. Advanced technologies, like AI, machine learning, and real-time data collection, are harnessed for precise risk assessments and prompt safety interventions.

The legal landscape, especially concerning data protection and worker rights, is indispensable. The General Data Protection Regulation (GDPR) necessitates strict adherence to principles of consent, transparency, data minimization, and rigorous security measures. Ensuring explicit permission, collecting only essential data, and safeguarding against unauthorized breaches are essential for successful and secure integration. Workers should be entitled to access, modify, and even erase their data, with extra rigor applied to international data transfers, particularly outside the European Economic Area (EEA). Beyond mere compliance, GDPR emphasizes proactive engagement with workers, addressing their concerns, and ensuring clear demarcation between personal and professional data realms. Additionally, the emerging principle of digital disconnection underscores the necessity to balance digital technology usage at work with overall employee well-being, promoting clear boundaries between professional obligations and personal life.

The ever-evolving landscape of technology necessitates continual research to enhance and perfect the presented solution, such as improving emotion recognition by speech tone analysis or increasing the accuracy of mental assessment by the current AI model, making them more empathetic to the workers. Also, the assessment of certain illnesses, for instance, early Parkinson’s Disease [13]. Furthermore, with the required security and protection, this Human Digital Twin can be a cornerstone of public health for governmental departments.

## Figures and Tables

**Figure 1 sensors-24-00655-f001:**
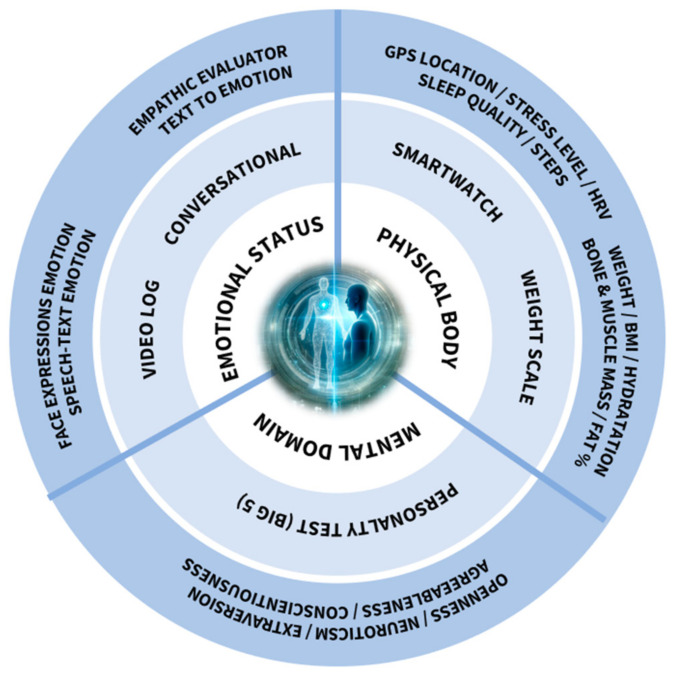
High-level architecture of the proposed HDT model.

**Figure 2 sensors-24-00655-f002:**
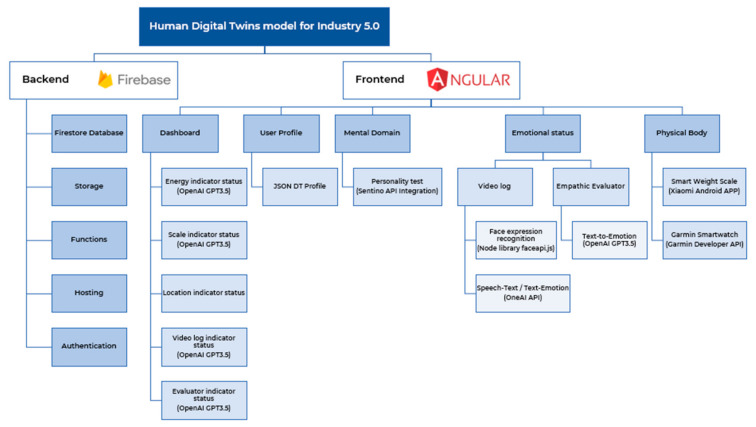
Detailed architectural diagram of frontend and backend framework.

**Figure 3 sensors-24-00655-f003:**
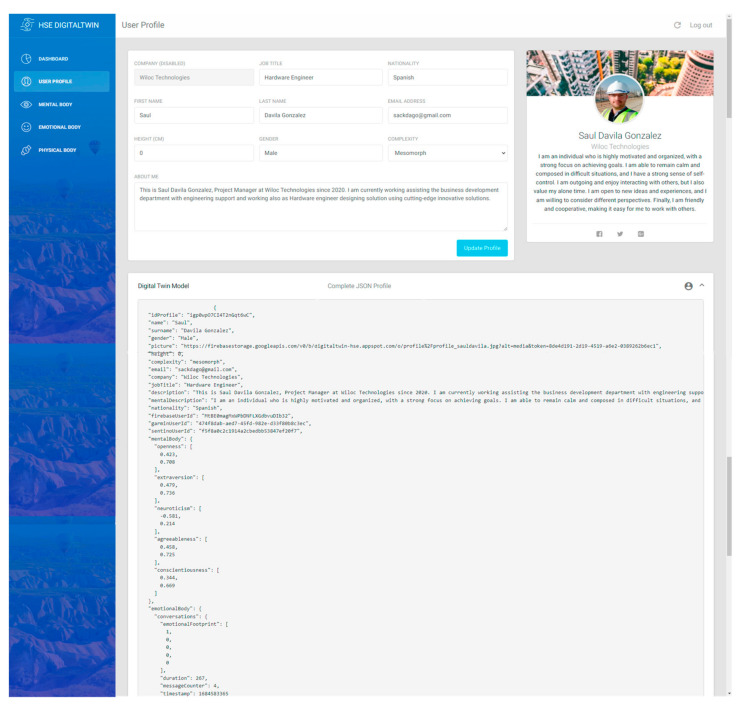
User profile view of the developed DT platform.

**Figure 4 sensors-24-00655-f004:**
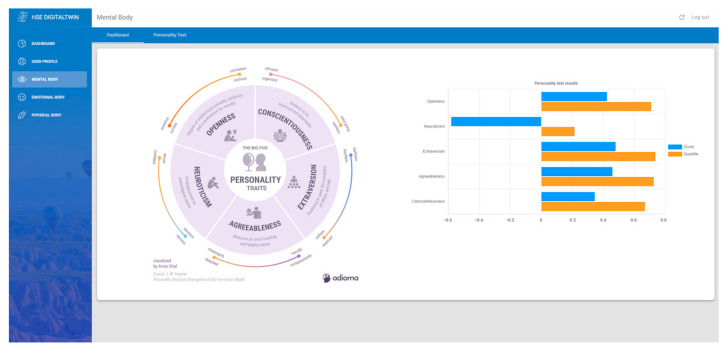
Dashboard of the mental body with personality analysis by Sentino API.

**Figure 5 sensors-24-00655-f005:**
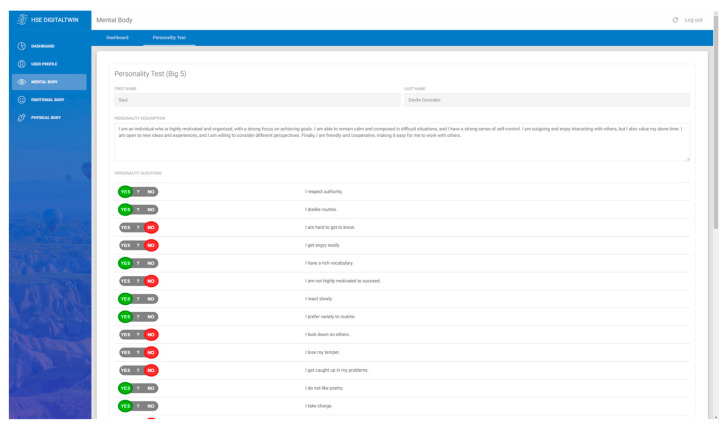
Big-Five personality test form.

**Figure 6 sensors-24-00655-f006:**
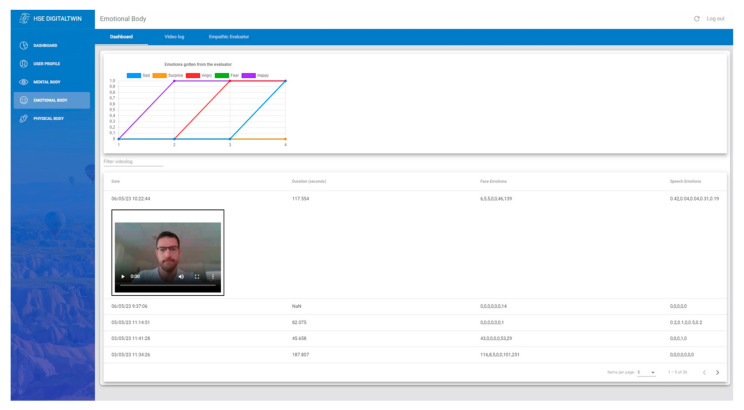
Dashboard of the emotional body of the developed DT platform.

**Figure 7 sensors-24-00655-f007:**
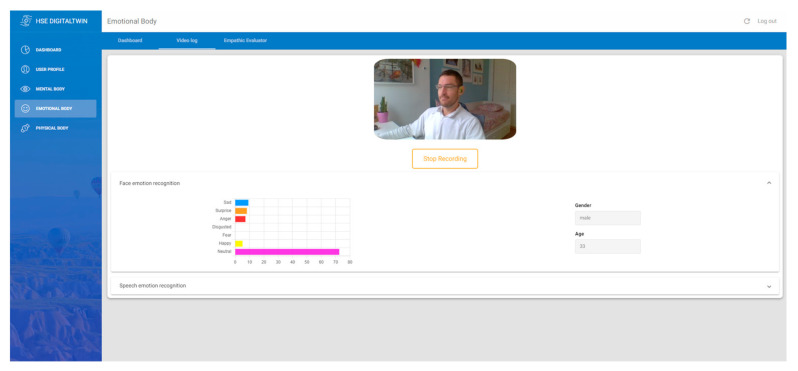
Video log to capture the facial expression through the webcam.

**Figure 8 sensors-24-00655-f008:**
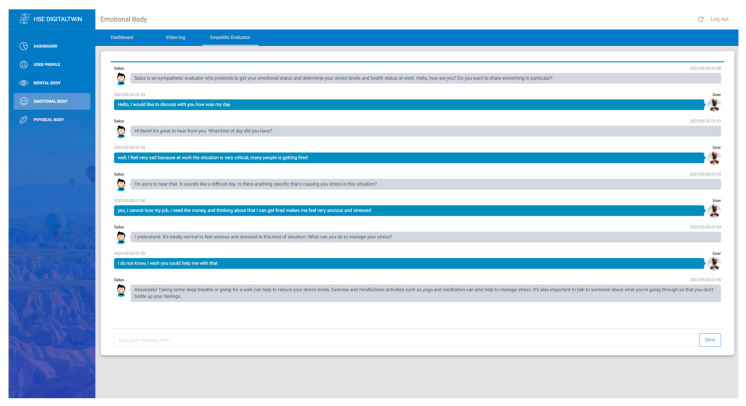
Emphatic evaluator based on Open AI GPT-3.

**Figure 9 sensors-24-00655-f009:**
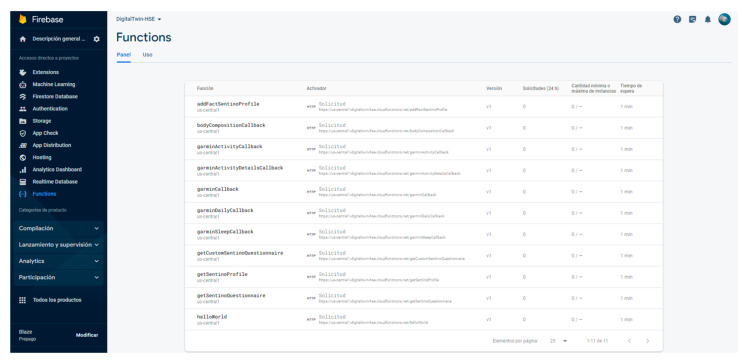
Google Firebase functions interface for interacting with third party application as services.

**Figure 10 sensors-24-00655-f010:**
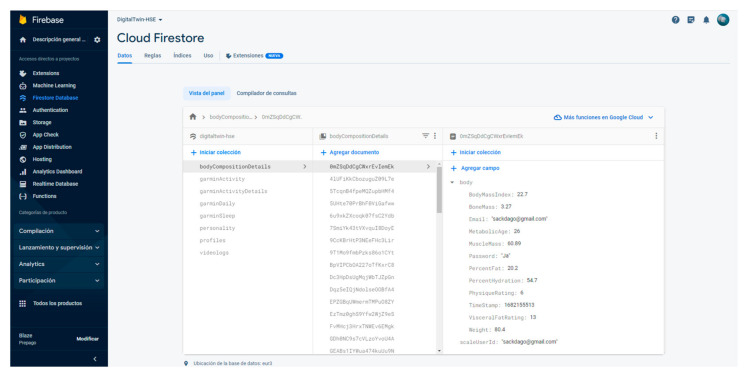
Google Firebase, Firestore database for storing worker data in non-relational databases.

**Figure 11 sensors-24-00655-f011:**
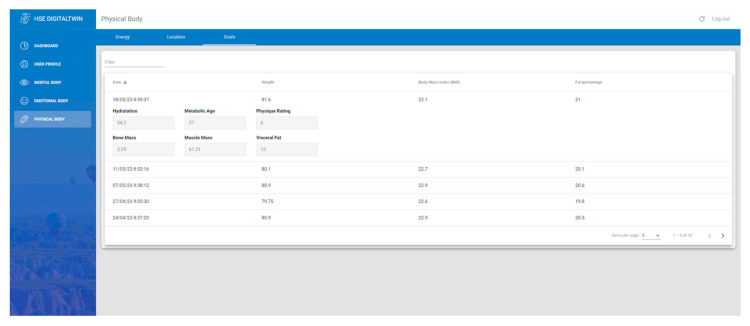
Body composition interface with the complete weight scale historical information.

**Figure 12 sensors-24-00655-f012:**
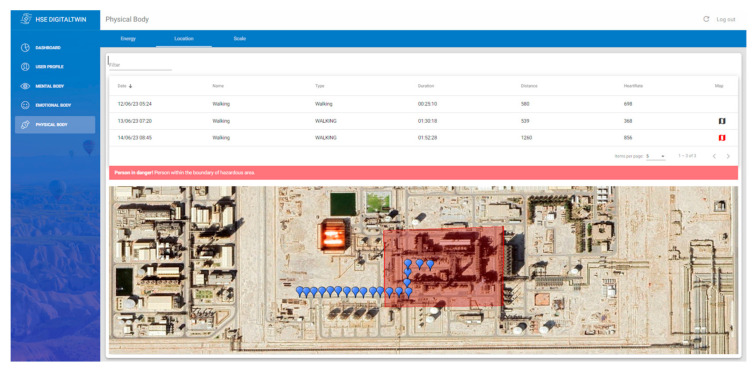
Physical location interface of the developed DT platform.

**Figure 13 sensors-24-00655-f013:**
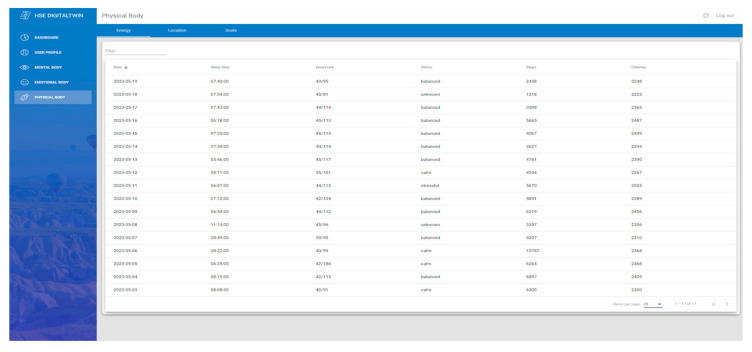
Table interface for the heart-related data with daily historical information.

**Figure 14 sensors-24-00655-f014:**
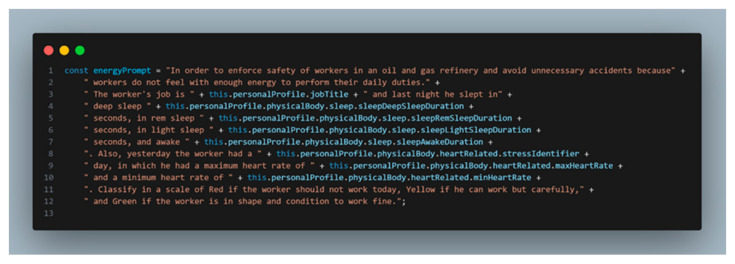
LLM prompt for obtaining and feeding the worker energy status indicator.

**Figure 15 sensors-24-00655-f015:**
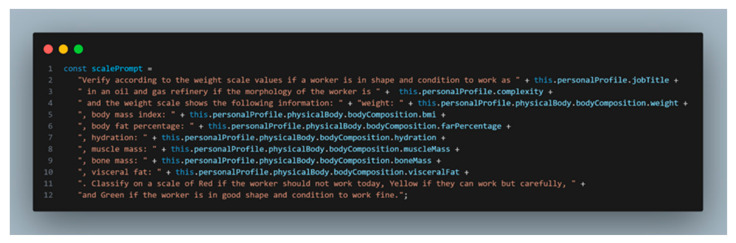
LLM prompt for obtaining and feeding the worker weight scale status indicator.

**Figure 16 sensors-24-00655-f016:**
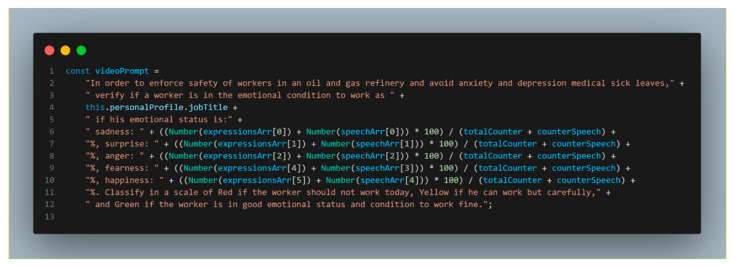
Worker video log AI indicator status.

**Figure 17 sensors-24-00655-f017:**
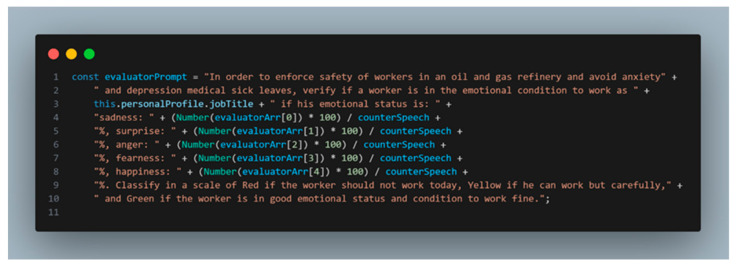
Worker chatbot evaluator AI indicator status.

**Figure 18 sensors-24-00655-f018:**
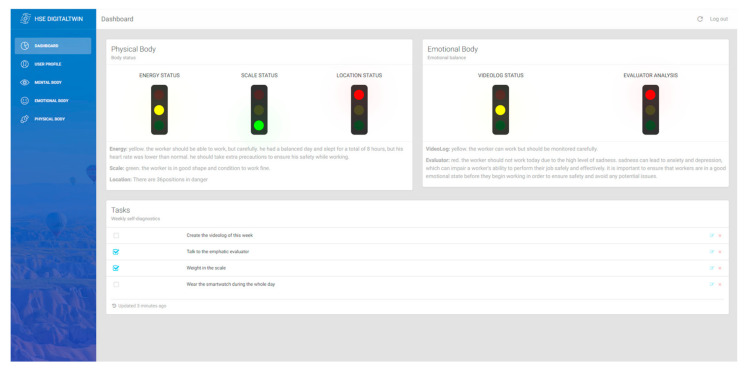
Dashboard of the developed DT platform.

**Table 1 sensors-24-00655-t001:** Comparative analysis of workers health and safety in publications: Focus on physical health, mental well-being, and emotional recognition.

Research	Year	Domain
Ref. [3] A safety management approach for Industry 5.0′s human-centered manufacturing based on digital twin	2023	Physical health
Ref. [5] Human digital twin system for operator safety and work management	2022	Physical health
Ref. [9] Enhancing Construction Safety Monitoring through the Application of Internet of Things and Wearable Sensing Devices	2019	Physical health
Ref. [12] Healthy Operator 4.0: A Human Cyber–Physical System Architecture for Smart Workplaces	2020	Physical health
Ref. [13] Using a smartwatch and smartphone to assess early Parkinson’s disease in the WATCH-PD study	2023	Physical health
Ref. [14] Wearable technology for the improvement of HSE management	2022	Physical health
Ref. [15] IoT-enabled model for Digital Twin of Mental Stress (DTMS)	2021	Mental well-being
Ref. [16] RT-PROFASY: Enhancing the well-being, safety, and productivity of workers by exploiting wearable sensors and artificial intelligence	2022	Mental well-being
Ref. [17] REDECA: A Novel Framework to Review Artificial Intelligence and Its Applications in Occupational Safety and Health	2021	Mental well-being
Ref. [18] Challenges of learning human digital twin: case study of mental wellbeing: Using sensor data and machine learning to create HDT	2023	Mental well-being
Ref. [19] Detection of stress, anxiety, and depression (SAD) in video surveillance using ResNet-101	2022	Mental well-being
Ref. [20] Artificial intelligence for chatbots in mental health: Opportunities and challenges	2021	Mental well-being
Ref. [21] Emotion Recognition for Affective Human Digital Twin by Means of Virtual Reality Enabling Technologies	2023	Emotional recognition
Ref. [22] Digital Twin Model: A Real-Time Emotion Recognition System for Personalized Healthcare	2022	Emotional recognition
Ref. [23] A Digital Twin-Driven Method for Product Performance Evaluation Based on Intelligent Psycho-Physiological Analysis	2021	Emotional recognition
Ref. [24] The Evaluation of Emotional Intelligence by the Analysis of Heart Rate Variability	2023	Emotional recognition
Ref. [25] Heart rate variability monitoring for emotion and disorders of emotion	2019	Emotional recognition

## Data Availability

Data are contained within the article.

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
