# Peer review of "Human Digital Twin in Industry 5.0: A Holistic Approach to Worker Safety and Well-Being through Advanced AI and Emotional Analytics"

_sensors, 2024, doi:10.3390/s24020655_

Round 1

Reviewer 1 Report

Comments and Suggestions for Authors

Although you address a relevant and interesting topic, it seems like there is improvement potential in presenting your concepts, the methods used as well as the results achieved. Your work seems very technology-oriented rather than research-oriented. You can properly define the Human Digital Twin from your perspective and present the the model you applied for it. Seeing how this is an IoT-related special issue, you can also include an architectural depiction of your framework, naming the interfaces and components you used to connect to different sensors and software components. 

The state of the art needs rework as you only mention work related to worker safety, do consider including previous work on the Human Digital Twin. 

Comments on the Quality of English Language

Please revise your use of capital words, which is irregular and often misplaced. 

Author Response

Date: 22-12-2023

MDPI Sensors

Response to Reviewers

Manuscript ID: sensors-2784172

Article Title: Human Digital Twin in Industry 5.0: A Holistic Approach to Worker Safety and Well-being through Advanced AI and Emotional Analytics

Dear Editor in Chief,

Thank you for allowing a resubmission of our manuscript, with an opportunity to address the reviewer’s comments and suggestions. We are uploading:

(a) our point-by-point response to the comments (below). 

(b) an updated manuscript with highlights indicating changes (Main Document).

Best regards,

Sergio Martin et al.

Author response:  Thanks for the opportunity of improving the manuscript again. We give responses and solutions to all the reviewers’ comments to ensure that the article is ready to be published.

AUTHORS' REPLY TO Reviewer 1.

Reviewer#1, Concern # 1: Although you address a relevant and interesting topic, it seems like there is improvement potential in presenting your concepts, the methods used as well as the results achieved. Your work seems very technology-oriented rather than research-oriented. You can properly define the Human Digital Twin from your perspective and present the model you applied for it.

Author response:  Thanks for the suggestion. We agree that it could enhance the manuscript.

Author action: We have extended section 3.1 with the definition and model of the Human Digital Twin proposed in the manuscript. Also, we have extended the introduction with the main contributions of this paper to the state-of-the-art.

Reviewer#1, Concern # 2: Seeing how this is an IoT-related special issue, you can also include an architectural depiction of your framework, naming the interfaces and components you used to connect to different sensors and software components.

Author response: Thanks for the suggestion. It can contribute to facilitating the understanding of the paper.

Author action: We have included a few high-level architectural images which describes the framework of this work and its interfaces.

Reviewer#1, Concern # 3: Please revise your use of capital words, which is irregular and often misplaced.

 Author response: Thanks for the suggestion.

Author action: We have reviewed the overall grammar and corrected the wrong capital words.

Reviewer 2 Report

Comments and Suggestions for Authors

Human digital twin is an important part of factory digital twin in Industry 5.0. The paper proposed some factors for HDT and developed a system to realize HDT. Some suggestions are as follows.  1. The main contributions of the paper should be clearly statemented. 2. There are many system interfaces in the paper. But the text is  unable to read. The text in the interface is more important that the interface style. 3. During the work, the worker usually do not speak. What is the type of the work in the paper ? 4. Fig10 shows a outside environment. Usually the work is in a factor floor. How about the positioning accuracy in a shop floor? 5. There is no a case study in the paper. Only the system functions are shown. How to realize the real time monitoring which is statemented in the abstract. 

Comments on the Quality of English Language

Moderate editing of English language required.

Author Response

Date: 22-12-2023

MDPI Sensors

Response to Reviewers

Manuscript ID: sensors-2784172

Article Title: Human Digital Twin in Industry 5.0: A Holistic Approach to Worker Safety and Well-being through Advanced AI and Emotional Analytics

Dear Editor in Chief,

Thank you for allowing a resubmission of our manuscript, with an opportunity to address the reviewer’s comments and suggestions. We are uploading:

(a) our point-by-point response to the comments (below). 

(b) an updated manuscript with highlights indicating changes (Main Document).

Best regards,

Sergio Martin et al.

Author response:  Thanks for the opportunity of improving the manuscript again. We give responses and solutions to all the reviewers’ comments to ensure that the article is ready to be published.

AUTHORS' REPLY TO Reviewer 2.

Reviewer#2, Concern # 1: The main contributions of the paper should be clearly statemented.

 Author response: Thanks for the suggestion.

Author action: We have extended section 1 with the main contributions of this paper to the state-of-the-art.

Reviewer#2, Concern # 2: There are many system interfaces in the paper. But the text is unable to read. The text in the interface is more important that the interface style.

 Author response:  Thanks for the suggestion.

Author action: We have replaced them with higher quality images.

Reviewer#2, Concern # 3: During the work, the worker usually do not speak. What is the type of the work in the paper ?

Author response: You are right, while working they do not speak. The work is mainly focused on those workers who are more susceptible to stress related issues, like supervisors and higher positions in an organization. The type of work is for Oil & Gas environments, an idea is that they can be forced to speak while working in the office, at the beginning or end of the day.

Author action: As per also the next concern, we have included in the introduction and in some other parts of the manuscript the potential scenario in which this manuscript is referring, Oil & Gas environments.

Reviewer#2, Concern # 4: Fig10 shows a outside environment. Usually, the work is in a factor floor. How about the positioning accuracy in a shop floor?

Author response: Thanks for the valuable suggestion. The article considers an outdoor scenario as an industrial environment, like an Oil & Gas construction plant, that is the reason GPS Location has been considered.

Author action: Due to the map view does not representing such environment, the image has been changed by another test in such environment. 

Reviewer#2, Concern # 5: There is no a case study in the paper. Only the system functions are shown. How to realize the real time monitoring which is statemented in the abstract.

Author response: Thank s for noticing this wrong idea. The paper aims to introduce a conceptual framework, not a case study.

Author action: We have modified the abstract to make sure readers do not get the wrong idea.

Reviewer#2, Concern # 6: Moderate editing of English language required.

Author response: Thanks for the suggestion.

Author action: We have reviewed the grammar and corrected.

Round 2

Reviewer 2 Report

Comments and Suggestions for Authors

It can be accepted.